# Data Exchange Markets via Utility Balancing

## ABSTRACT

This paper explores the design of a balanced data-sharing market-place for entities with heterogeneous datasets and machine learning models that they seek to refine using data from other agents. The goal of the marketplace is to encourage participation for data sharing in the presence of such heterogeneity. Our market design approach for data sharing focuses on interim utility balance, where participants contribute and receive equitable utility from refinement of their models. We present such a market model for which we study computational complexity, solution existence, and approximation algorithms for welfare maximization and core stability. We finally support our theoretical insights with simulations on a mean estimation task inspired by road traffic delay estimation.

**ACM Reference Format:**
Anonymous Author(s). 2024. Data Exchange Markets via Utility Balancing. In *Proceedings of TheWebConf 2024 (WWW 2024)*. ACM, New York, NY, USA, 14 pages. https://doi.org/XXXXXXX.XXXXXXX

## 1 INTRODUCTION

The power of big data comes from the improved decision making it enables via training and refining machine learning models. To unlock this power to the fullest, it is critical to enable and facilitate data sharing among different units in an organization and between different organizations. The market for big data "*accounted for USD 163.5 Billion in 2021 and is projected to occupy a market size of USD 473.6 Billion by 2030 growing at a CAGR of 12.7%*" [29]. Motivated by the emergence of online marketplaces for data such as SnowFlake [11], in this paper we consider the timely question:

> How can we design a principled marketplace for sharing data between entities (organizations or applications) with heterogeneous datasets they own and machine learning models they seek to refine, so that each entity is encouraged to voluntarily participate?

Towards this end, we assume agents have diverse ML models for decision making that they seek to refine with data. At the same time, each agent possesses data that may be relevant to the tasks of other agents. As an example, a retailer may have sales data for certain products in certain geographic locations, but may want data for related products in other markets to make a better prediction of sales trends. This data could be in the hands of competing retailers. Similarly, a hospital system seeking to build its in-house model for

a disease condition based on potentially idiosyncratic variables may want patient data from other hospital systems to refine this model.

In our paper, we assume the participants in the market have no value for money. We further assume that the agents seeking data are the same as those seeking to refine models. Therefore we consider an exchange economy without money as opposed to a two-sided market with buyers and sellers. This is a reasonable assumption for non-profits such as hospital systems or universities, where student or patient data can be "exchanged" but not sold for profit. Though we seek a market design without money, the agents in the market still need to be incentivized to voluntarily participate in the market and exchange data, and this is the main focus of our design.

In the settings we consider, data is often sensitive and private [3, 16]. As in [3], we address this issue by having a trusted central entity (or clearinghouse) with who all agents share their ML tasks and datasets. This entity can refine or retrain the model for one agent using samples of the data from other agents. For instance, if each agent specifies the gradient of their loss function and their in-house model parameters, the central entity can run stochastic gradient descent to update the parameters using the other data. This way, the central entity can efficiently compute the loss of the refined model and hence the utility of a collection of datasets to a model. By using a utility sharing method such as Shapley value that has been well-studied in machine learning [17, 18], the entity can use the same process to attribute this utility gain fairly to the agents that contributed data to the refinement. The entity then sends the refined models back to the respective agents, preserving data privacy in the process.

## 1.1 Model and Results

Our approach to market design for data exchange without money is to view it as *utility balancing* – to encourage voluntary participation, an agent should contribute as much utility to other agents as they receive from them. In market design terminology, this corresponds to having a common endogenous price per unit utility bought or sold, so that each agent is revenue-neutral. The goal of the central entity is to find the right amount of data any set of agents should exchange, so that the overall solution is utility balanced. The solution is randomized, where for each agent, we compute a distribution over sets of other agents. When this agent chooses a set from this distribution to obtain data from, then utility balance holds in expectation (or *interim*). We motivate interim balance in settings where the same agents trade over many epochs so that the total utility across these epochs approaches its expectation. The objective of the central entity could be to either maximize total utility of agents (social welfare maximization), or core-stability among coalitions of agents.

We call this overall problem the DATA EXCHANGE PROBLEM. We study computational complexity and existence results for the DATA EXCHANGE PROBLEM under natural utility functions and how that utility is shared among contributors. At a high level, our main results are the following.

(1) We present a formal model for the DATA EXCHANGE PROBLEM in Section 2 based on interim utility balancing, codifying the objectives of welfare maximization and stability.

(2) We show NP-HARDNESS (Appendix B) and polynomial time approximation algorithms for welfare maximization (Section 3 and Appendix C). We present a logarithmic approximation in Theorem 6 for submodular utilities and a general class of sharing rules that includes the well-known Shapley value, and a PTAS for concave utilities with proportional sharing in Theorems 6 and 7.

(3) We show the existence of core stable and strategyproof solutions and the trade-offs achievable between these notions and welfare in Appendix D. We also show that a specific type of stable and strategyproof solution can be efficiently computed via greedy matchings.

(4) We finally perform simulations on a road network where agents are paths that are interested in minimizing sample variance. We show that our approximation algorithms significantly outperform a pairwise trade benchmark, showing the efficacy of our model and algorithms.

We present the statements of these results more formally in Section 2 after we present the formal mathematical model.

## 1.2 Related Work

The emerging field of data markets already has unearthed several novel challenges in data privacy, market design, strategyproofness, and so on. Please see recent work [3, 16] for a comprehensive enumeration of research challenges. Our paper makes progress on some of these challenges by proposing a market design via a central clearinghouse and utility balancing, with computational and stability analysis.

*Exchange Economies.* Our paper falls in the framework of *market design.* Though market design for exchange economies – where agents voluntarily participate in trade given their utility functions and the market constraints – is a classic problem, much of this work concerns markets for goods that cannot be freely replicated. The key challenge in our setting is that data can be freely replicated, which makes the market design problem very different.

There are two classic exchange economies that relate to our work – the trading of indivisible goods [32] and market clearing [5]. The first classic problem is also termed the *house allocation problem.* Here, every agent owns a house and has a preference ordering over other houses. The goal is to allocate a house to each agent in a fashion that lies in the core: No subset of agents can trade houses and improve their outcome. Shapley and Scarf [32] showed that the elegant *top trading cycles* algorithm finds such a core-stable allocation. A practical application of this framework is to *kidney exchanges* [30], which is widely studied and implemented.

Our problem falls in the same framework as house allocation, albeit with data instead of houses. Data is a replicable resource, and leads to complex utilities for agents; these aspects make the algorithm design problem very different, as we compare in Appendix A. We further note that strategyproofness is a big consideration in kidney exchanges [6, 30], for instance, hospitals can be incentivized to match patient pairs internally and not participate in inter-hospital exchanges. Similar issues could arise in data exchange. Though

strategyproofness is not the main focus of our paper, we present results for it in Appendix D.4.

In the same vein, the second classic problem of market clearing for non-replicable goods dates back to Arrow and Debreau [5], and has elegant solutions via equilibrium pricing of the goods. However, equilibrium prices are harder to come by for *replicable* digital goods such as music or video [23]. We bypass this issue by having a common price per unit of utility traded, which translates, via eliminating the price, to our flow formulation on utilities.

*Federated Learning.* Our work is closely related to recent work by Donahue and Kleinberg [14, 15] on forming coalitions for data exchange in federated learning. However, in their settings, all agents have the same learning objective (either regression or mean estimation), but have data with different bias, leading to local models with different bias. The goal is to form coalitions where the error of the model for individual agents, measured against their own data distribution, is minimized. The authors present optimal coalitional structures for maximizing welfare, as well as achieving core stability. In contrast, we consider agents with *heterogeneous* tasks and data requirements, which makes even welfare maximization NP-HARD (without considering core stability).

*Pricing and Shapley Value.* In the settings we study, agents are both producers and consumers of data, motivating an exchange economy like the works cited above. When sellers of data are distinct from buyers, various works [3, 8, 10, 12, 19] have studied pricing and incentives for selling aspects such as privacy and accuracy. See [27] for a survey.

One important aspect of our work is allocating utility shares to the agents contributing data. For most of our paper, we adopt the Shapley value [33]. Though this method has its roots in cost sharing in Economics, it has seen a resurgence in interest as a method to measure the utility of individual datasets for a machine learning task [17, 18]. This method has many nice properties; see [3] for a discussion of these properties in a data sharing context. We note that our work presents a general framework and as we show in the paper, it can be adapted to other utility sharing rules.

## 2 THE DATA EXCHANGE PROBLEM AND OUR RESULTS

Without further ado, we formally present the DATA EXCHANGE PROBLEM and a summary of our results. We are given a set of agents $X$. Each agent $i \in X$ has a dataset $\mathcal{D}_i$ and a machine learning task $t_i$. (Our results easily extend to the setting where each agent has multiple datasets and tasks.) The accuracy of the task $t_i$ can be improved if agent $i$ obtains the datasets of other users.

### 2.1 Utility Functions

Suppose agent $i$ obtains the datasets $\cup_{j \in S} \mathcal{D}_j$ of a subset $S$ of agents, then the improvement in accuracy is captured by a *utility function* $u_i(S)$. We assume this function can be computed efficiently for a given set $S$ of agents. Further, this set function is assumed to satisfy the following:

**Non-negativity and Boundedness:** $u_i(S) \in [0, 1]$ for all $S \subseteq X \setminus \{i\}$. Furthermore, $u_i(\emptyset) = 0$. By scaling, we can also assume that $\max_i u_i(X) = 1$.

**Monotonicity:** $u_i(S) \geq u_i(T)$ for all $T \subset S$.

**Submodularity:** This captures diminishing returns from obtaining more data. For all $T \subset S$ and $q \notin S$, we have
$$u_i(S \cup \{q\}) - u_i(S) \leq u_i(T \cup \{q\}) - u_i(T).$$

A special case of submodular utilities is the **symmetric weighted** setting: Here, there is a concave non-decreasing function $f_i$ for each agent $i$. Suppose agent $j$'s dataset that she contributes to $i$ has size $s_{ij}$, then we have $u_i(S) = f_i\left(\sum_{j \in S} s_{ij}\right)$. In other words, the utility only depends on the total size of the datasets contributed by the agents in $S$.

EXAMPLE 1. *Suppose each agent $i$ is interested in estimating the population mean of data in its geographical vicinity, and its utility function is the improvement in variance of this estimate. In this case, agent $j$ can contribute $s_{ij}$ amount of data to agent $i$, and we let $D_i(S) = \sum_{j \in S} s_{ij}$. Assuming $s_{ii} = 1$ and that these data are drawn i.i.d. from a population with variance $\sigma_i^2$, we have $u_i(S) = \sigma_i^2 \left(1 - \frac{1}{1 + D_i(S)}\right)$ and falls in the symmetric weighted setting.*

*Continuous Utilities.* Though the bulk of the paper focuses on utilities modeled as set functions, in Appendix C, we also consider the setting where agents can exchange fractions of data. Suppose agent $j$ transfers $y_{ij}$ fraction of her data to agent $i$, then agent $i$'s utility is modeled as a continuous, monotonically non-decreasing function $u_i(\vec{y}_i) \in [0, 1]$, where $\vec{y}_i = \langle y_{i1}, y_{i2}, \ldots \rangle$. As we show later, such utilities lead to more tractable algorithmic formulations.

## 2.2 Utility Sharing

The utility $u_i(S)$ that $i$ gains from the set $S$ of agents is attributed to the agents in $S$ according to a fixed rule. We let $h_{ij}(S)$ denote the contribution of agent $j \in S$ to the utility $u_i(S)$, so that $\sum_{j \in S} h_{ij}(S) = u_i(S)$. In this paper, we consider two classes of sharing rules that have been studied in cooperative game theory, and more recently in machine learning:

**Shapley Value:** This is a classic "gold-standard" rule from cooperative game theory [17, 18, 33], and works as follows: Take a random permutation of the agents in $S$. Start with $W$ as the empty set and consider adding the agents in $S$ one at a time to $W$. At the point where $j$ is added, let $\Delta_j = u_i(W \cup \{j\}) - u_i(W)$ be the increase in utility due to the datasets in $W$. The Shapley value $h_{ij}(S)$ is the expectation of $\Delta_j$ over all random permutations of $S$.

**Proportional Value:** In this class of rules [9, 24], there is a fixed set of weights $\{w_{ij}\}$, and we define $h_{ij}(S) = \frac{w_{ij}}{\sum_{k \in S} w_{ik}} \cdot u_i(S)$. The natural special case is the setting $w_{ij} = u_i(\{j\})$, so that the utility is shared proportionally to how much $j$'s dataset would have individually contributed to $i$.

For submodular utilities, the Shapley value satisfies a property called *cross-monotonicity* [25]: if $T \subset S$ and $j \in T$, then $h_{ij}(T) \geq h_{ij}(S)$. Note that there is an entire class of rules that satisfy cross-monotonicity for submodular utilities; please see [17, 18] for a detailed discussion of the Shapley value and related cross-monotonic rules in the context of machine learning. In contrast, the proportional value does not satisfy this property. We contrast these rules in the following example.

EXAMPLE 2. *There are $n$ agents each contributing data to agent 0. The first $n - 1$ agents have identical data, so that $u_0(S) = 0.5$ for any non-empty $S \subseteq [n - 1]$. Agent $n$ has a unique dataset so that $u_0(\{n\}) = 0.5$, and $u_0(S \cup \{n\}) = 1$ for any non-empty $S \subseteq [n - 1]$. Then, for $S \subseteq [n - 1]$, the Shapley value is $h_{0n}(S \cup \{n\}) = 0.5$ and $h_{0j}(S \cup \{n\}) = \frac{1}{2|S|}$ for $j \in S$. However, the proportional share with $w_{ij} = u_i(\{j\})$ is $h_{0j}(S \cup \{n\}) = \frac{1}{|S|+1}$ for all $j \in S \cup \{n\}$.*

In the above example, the Shapley value is more reflective of the actual contributions of the individual agents compared to proportional value; however, the latter rule sometimes leads to better algorithmic results. In particular, for continuous concave utilities and the symmetric weighted setting, the proportional sharing rule is more tractable, while for general submodular utilities, the Shapley value is more tractable.

EXAMPLE 3. *For the symmetric weighted setting described above, the proportional value with $w_{ij} = s_{ij}$ has a relatively simple form: $h_{ij}(S) = \frac{s_{ij}}{\sum_{k \in S} s_{ik}} \cdot f_i(\sum_{k \in S} s_{ik})$.*

## 2.3 Constraints for DATA EXCHANGE: Utility Flow

We now present the constraints of the DATA EXCHANGE PROBLEM. We assume there is a central entity that computes this exchange. The key constraint is that each agent receives as much utility from the exchange as it contributes. In this exchange, each agent $i$ is associated with a distribution $\{x_{iS}\}$ over sets $S \subseteq X \setminus \{i\}$ of agents whose datasets she could receive. In other words, with mutually exclusive probability $x_{iS}$, agent $i$ receives the datasets from $S$ and receives utility $u_{iS}$ as a result.

The first constraint encodes that $\{x_{iS}\}$ define a probability distribution over possible sets $S$.

$$\forall i, \sum_S x_{iS} \leq 1 \qquad (1)$$

where the remaining probability is assigned to $S = \emptyset$.

The BALANCE condition captures that the expected utility contributed by an agent to other agents is equal to the expected utility she receives.

$$\forall i, \sum_S \sum_{j \in S} h_{ij}(S) x_{iS} = \sum_j \sum_{S | i \in S} h_{ji}(S) x_{jS} \qquad (2)$$

Note that the balance condition is *interim*, meaning it holds for the expected utility. Any solution that satisfies the BALANCE condition subject to Eq. (1) is said to be a *feasible* solution to the DATA EXCHANGE PROBLEM.

EXAMPLE 4. *The above model has an interesting connection to Markov chains. Suppose we restrict $S$ to either be $\emptyset$ or $S_i = X \setminus \{i\}$. Let $v_{ij} = h_{ij}(S_i)$ and $y_i = x_{iS_i}$. Then we have the constraints:*

$$y_i \cdot \sum_{j \in S_i} v_{ij} = \sum_{j | i \in S_j} y_j v_{ji} \qquad \forall i \in X$$

$$y_i \in [0, 1] \qquad \forall i \in X$$

*Set $w_i = \sum_{j \in X \setminus \{i\}} v_{ij}$, and $p_{ij} = \frac{v_{ij}}{w_i} \in [0, 1]$. Then, $\sum_{j \in S_i} p_{ij} = 1$ for all $i$. Further, setting $z_i = y_i w_i$, the first constraint becomes*

$$z_i = \sum_{j | i \in S_j} z_j p_{ji}.$$

*Then, treating the $p_{ji}$ as transition probabilities from $j$ to $i$ in a Markov chain, the $\{z_i\}$ are the steady state probabilities of the chain. Assuming all $p_{ji} > 0$, by the Perron-Frobenius theorem, there is a unique set of non-negative values $\{z_i\}$.*

**Remarks.** First note that for deterministic exchange where $x_{iS} \in \{0, 1\}$, the balance constraints may not have a feasible solution. This motivates our use of randomization and interim balance. A randomized solution is justified when agents interact over many epochs with different datasets and models. Though any specific interaction is ex-post imbalanced, these even out over time by the law of large numbers. Such interim balance also makes our algorithmic problem more tractable.

Next, though we don't discuss it in the paper, it is easy to generalize the model to the setting where each agent $i$ has a collection of datasets and a collection of tasks, and each needs different datasets. Further, in Appendix C, we discuss the changes that need to be made to the constraints to handle continuous, concave utilities.

Finally, as mentioned before, we assume the clearinghouse has accurate access to all datasets and tasks, and can hence compute utilities, their shares, and the feasible DATA EXCHANGE solution. We ignore strategic misreporting on the part of the agents for most of the paper, but we will discuss this aspect and its trade-off with other objectives towards the end in Appendix D.4.

## 2.4 Social Welfare Objective

Our goal is to find the optimal DATA EXCHANGE subject to feasibility. Towards this end, we mainly consider the *social welfare* objective where the goal is to find the distributions $\{x_{iS}\}$ that maximizes:

$$\text{Social Welfare} = \sum_{i \in X} \sum_{S \subseteq X \setminus \{i\}} u_i(S) x_{iS} = \sum_{i \in X} \sum_{S \subseteq X \setminus \{i\}} \sum_{j \in S} x_{iS} h_{ij}(S). \quad (3)$$

We will study the computational complexity of this problem.

**Remark about running times.** Throughout, we assume that there is an efficient subroutine MLSUB that given an agent $i$ and set $S \subseteq X \setminus \{i\}$ returns the utility $u_i(S)$ and the shares $h_{ij}(S)$ for all $j \in S$. We remark that by "polynomial" running time, we mean polynomially many calls to MLSUB, combined with polynomially many ancillary computations. Such an approach decouples the exact running time of MLSUB from our results. For ML tasks, estimating $u_i(S)$ will require retraining the model using data from $S$; this can typically be done efficiently. Further, estimate $h_{ij}(S)$ can be done to a good approximation via sampling permutations; see [3, 18].

**Computational complexity of welfare maximization.** The welfare maximization problem is a linear program with $2n$ constraints, so that the optimum solution has at most $2n$ non-zero variables. Nevertheless, we show NP-HARDNESS by a reduction from EXACT 3-COVER. We note that the hardness result holds even when for any $S$, both $u_i(S)$ and $h_{ij}(S)$ are computable in near-linear time.

THEOREM 5 (PROVED IN APPENDIX B). *The welfare maximization objective in* DATA EXCHANGE *is* NP-HARD *for submodular utilities and Shapley value sharing.*

In Section 3, we develop polynomial time algorithms that multiplicatively approximate social welfare.[1] Our algorithms achieve approximate feasibility, where we relax the BALANCE constraint to $\epsilon$-BALANCE (where $\epsilon \in (0, 1)$):

$$\left| \sum_S \sum_{j \in S} h_{ij}(S) x_{iS} - \sum_j \sum_{S | i \in S} h_{ji}(S) x_{jS} \right| \leq \epsilon \qquad \forall i. \quad (4)$$

The running times we achieve are now polynomial in $\frac{1}{\epsilon}$, with the assumption that there are analogously many calls to MLSUB. We show the following theorem in Section 3; the precise running time and approximation factors are presented there.

THEOREM 6 (PROVED IN SECTION 3). *We can achieve the following approximation factors to the social welfare objective for* DATA EXCHANGE *via an algorithm that runs in time polynomial in the input size and $\frac{1}{\epsilon}$ and finds a feasible solution that satisfies $\epsilon$-BALANCE:*
- *A $O(\log n)$ approximation for arbitrary submodular utilities[2] and any cross-monotonic utility sharing rule (including the Shapley value rule).*
- *A $1 + \epsilon$ approximation for symmetric weighted setting and proportional value with $w_{ij} = s_{ij}$.*

Our results follow by writing the social welfare optimization problem as a Linear Program (LP) with exponentially many variables of the form $\{x_{iS}\}$. Since the number of feasibility constraints is $2n$, we use the multiplicative weight method to approximately solve it. This requires developing a dual oracle for the constraints, which for each agent $i$, is a constrained maximization problem over a weighted sum of $\{h_{ij}(S)\}$, and we need to find the set $S \subseteq X \setminus \{i\}$ that maximizes this weighted sum. We show approximation algorithms for this problem, leading to the proof of the above theorem.

Further, in Appendix C, we show the following theorem (see Appendix C for the formal model):

THEOREM 7 (PROVED IN APPENDIX C). *For* DATA EXCHANGE *with continuous concave utility functions and proportional sharing, for any $\epsilon \in (0, 1)$, there is an algorithm running in time polynomial in the input size and $\frac{1}{\epsilon}$ and that finds a $(1 + \epsilon)$ approximation to social welfare, while violating BALANCE by an additive $\epsilon$.*

## 2.5 Core Stability and Strategyproofness

Stability is a widely studied notion in cooperative game theory, and seeks solutions that are robust to coalitional deviations. In our context, we have the following definition.

DEFINITION 8. *A feasible solution $\mathcal{F}$ to* DATA EXCHANGE *is core stable if there is no $U \subseteq X$ of users and another feasible solution $\mathcal{F}'$ just on the users in $U$ such that for all $i \in U$, $u_i(\mathcal{F}') > u_i(\mathcal{F})$. A solution $\mathcal{F}$ is c-stable if there is no such $U$ with $|U| \leq c$.*

In other words, suppose a coalition $U \subseteq X$ of agents deviates and trades just among themselves via a feasible solution $\mathcal{F}'$ so that all their utilities improve, then this coalition is *blocking*. A core solution has no blocking coalitions.

---

[1]By $\alpha$-approximation for $\alpha \geq 1$, we mean our algorithm achieves at least $\frac{1}{\alpha}$ fraction of the optimal social welfare.
[2]The results hold for arbitrary monotone utilities and only require cross-monotonic sharing; however, cross-monotonicity typically does not hold unless utilities are submodular.

In Appendix D, we first show that regardless of the utility function and choice of sharing rule, there is always a feasible DATA EXCHANGE solution that is core-stable to an arbitrarily good approximation (Theorem 18). This is a consequence of Scarf's lemma [31] from cooperative game theory. Though it is unclear how to efficiently compute such a solution in general, we show an algorithm to find a 2-stable solution via GREEDY maximal weight matching.

We next study the trade-off between core and welfare. On the negative side, we show an instance in the symmetric weighted setting with proportional sharing, where any core solution has social welfare that is $\Omega(\sqrt{n})$ times smaller than the optimal social welfare (Theorem 20), showing the two concepts of core and welfare maximization can be far from each other. Nevertheless, we show (Theorem 21) how to achieve approximate core-stability and social welfare simultaneously via randomizing between them.

We finally consider strategic behavior by agents, where they hide either their tasks or data. We define feasible misreports in Appendix D.4, and again show that for the symmetric weighted setting, strategyproofness and approximate welfare maximization are simultaneously incompatible (Theorem 23). On the positive side, we show that a GREEDY cycle canceling algorithm that generalizes greedy matching is strategyproof.

*Comparison to kidney exchange.* We present a comparison of our results above with kidney exchange in Appendix A.

## 3 ALGORITHMS FOR WELFARE MAXIMIZATION: PROOF OF THEOREM 6

In this section, we present approximation algorithms for welfare maximization. We present the overall framework in Section 3.1, which reduces the problem to solving an oracle problem, one for each agent (Eq. (10)), so that an approximation algorithm to the oracle translates to the same approximation to welfare maximization, while achieving $\epsilon$-balance (Eq. (4)). We present the approximations to the oracle for submodular utilities with Shapley value in Section 3.3, and for symmetric weighted concave utilities with proportional sharing in Section 3.4. We present an extension to continuous concave utilities with proportional sharing in Appendix C.

As mentioned before, the welfare maximization problem can be written as an exponential-sized LP, where the non-negative variables are $\{x_{iS}\}$; the objective is to maximize Eq. (3) subject to the constraints Eqs. (1) and (2).

### 3.1 Multiplicative Weight Algorithm

We solve this using the multiplicative weights framework of Plotkin, Shmoys, and Tardos (PST) [28]. Since our final solution loses an additive $\epsilon$ in the balance constraints (Eq. (2)), we assume at the outset that these constraints are violated by an additive $\epsilon$, that is, Eq. (4). The problem with relaxed constraints can only have a larger objective value (social welfare). The relaxation helps us achieve polynomial running time.

LEMMA 9. *Let OPT denote the optimal solution value to the instance with relaxed balance constraints. Then $OPT \geq \epsilon$.*

PROOF. To see this, recall that we assumed $\max_i u_i(X) = 1$. For the maximizer $i$, set $x_{iX} = \epsilon$ and set all other variables to zero. This gives us a guarantee that $OPT \geq \epsilon$. □

Now, we try all objective values in powers of $(1+\epsilon)$ using binary search. Consider some guess $B$ for this value; we want to check if this value is feasible. By Lemma 9 we assume that $B \geq \epsilon$. We therefore want to check the feasibility of the following LP, where the objective Eq. (3) is encoded in Eq. (5); the balance constraints Eq. (4) is encoded in Eqs. (6) and (7); and the probability constraint Eq. (1) is encoded in Eq. (8). Call this LP1($B, \epsilon$). Our final solution will correspond to the largest $B$ for which LP1($B, \epsilon$) is feasible.

$$\sum_{i,j,S} h_{ij}(S)x_{iS} \geq B \qquad (5)$$

$$\forall i, \sum_{j,S} h_{ij}(S)x_{iS} - \sum_{j,S|i\in S} h_{ji}(S)x_{jS} \geq -\epsilon \qquad (6)$$

(LP1)
$$\forall i, -\sum_{j,S} h_{ij}(S)x_{iS} + \sum_{j,S|i\in S} h_{ji}(S)x_{jS} \geq -\epsilon \qquad (7)$$

$$\forall i, \sum_{S} x_{iS} \leq 1 \qquad (8)$$

$$\forall i, S, \sum_{S} x_{iS} \geq 0 \qquad (9)$$

We will use the PST framework to solve the feasibility of the above LP. Let Eqs. (5) to (7) be represented by the coefficient matrices $A, b$ and let $P$ be the polytope of vectors satisfying Eqs. (8) and (9). We are testing whether $\exists?x \in P, Ax \geq b$. The PST framework requires an oracle to solve $\max_{x \in P} p^\top Ax$ for arbitrary vectors $p$. In our setting, this becomes

$$\text{Oracle} = \max_{x \in P} \sum_{i,j,S} Q_{ij}h_{ij}(S)x_{iS}$$

for possibly negative weights $Q_{ij}$. Since the constraints across $i$ are now independent, the maximum solution will select the optimum solution $S$ to Eq. (10) and sets $x_{iS} = 1$, for each $i$.

$$\text{Oracle for agent } i = \max_{S} \sum_{j\in S} Q_{ij}h_{ij}(S) \qquad (10)$$

Using a similar proof as Theorem 5, it can be shown the Oracle problem is NP-HARD. We will therefore develop approximation algorithms, and show two such algorithms in Sections 3.3 and 3.4. As we show below, this will translate to an approximation for the social welfare. The overall algorithm is presented in Algorithm 1.

### 3.2 Analysis

Suppose the multiplicative approximation ratio of the oracle Eq. (10) is $\alpha \geq 1$; this means the oracle subroutine finds a solution whose value is at least $OPT/\alpha$ when $OPT$ is the optimal solution to the oracle. Define $\rho$ be the maximum value that any of the constraints in $Ax \geq b, x \in P$ can be additively violated. Since we assume $u_i(X) \leq 1$ for all $i$, it is clear that $\rho = \sum_i u_i(X) \leq n$.

Our main theorem is the following.

THEOREM 10. *Suppose the oracle problem Eq. (10) can be solved to a multiplicative approximation factor of $\alpha$. Then, with $O(\frac{n^2\alpha^2 \log n}{\epsilon^2})$ calls to the oracle subproblem and $O(n)$ time overhead per call to the oracle, Algorithm 1 returns a solution $\mathbf{x}$ that satisfies Eqs. (6) to (9) and that satisfies:*

$$\sum_{i,j,S} h_{ij}(S)x_{iS} \geq \frac{OPT}{2\alpha(1+3\delta)}.$$

---

**Algorithm 1** Multiplicative Weights Update to solve LP1.

---

1: Choose parameters $\epsilon, \delta \leq 1$ and $\eta = \frac{\epsilon}{4n\alpha}$.
2: Try values for $B$ via in powers of $(1 + \delta)$.
3: Let $A \in \mathbb{R}^{(2n+1) \times n}, b \in \mathbb{R}^{2n+1}$ denote the coefficients of LP1$(B, \epsilon)$.
4: Let $\mathbf{w}^{(1)} = \mathbf{1}^{2n+1}$.
5: **for** $t = 1, \ldots, T = \frac{32n^2\alpha^2 \log n}{\epsilon^2}$ **do**
6:      Let $\mathbf{p}^{(t)} := \frac{\mathbf{w}^{(t)}}{\sum_i w_i^{(t)}}$.
7:      Let $\mathbf{x}^{(t)}$ be the output of the $\alpha$-approximate oracle with input $\mathbf{p}^{(t)\top} A\mathbf{x}$.
8:      **if** $\mathbf{p}^{(t)\top} A\mathbf{x}^{(t)} < \mathbf{p}^{(t)\top} \frac{b}{\alpha}$ **then**
9:          Return infeasible and decrease the guess for $B$.
10:      **else**
11:          $\mathbf{m}^{(t)} := \frac{1}{\rho}(A\mathbf{x}^{(t)} - \frac{b}{\alpha})$.
12:          $\forall i, w_i^{(t+1)} := w_i^{(t)}(1 - \eta m_t^{(t)})$.
13:      **end if**
14: **end for**
15: Return $\bar{\mathbf{x}} = \frac{\sum_t \mathbf{x}^{(t)}}{T}$.

---

To prove this theorem, we require a result from [4].

LEMMA 11 (THEOREM 2.1 IN [4]). *After $T$ rounds in Algorithm 1, for every $i$,*

$$\sum_{t=1}^{T} \mathbf{m}^{(t)} \cdot \mathbf{p}^{(t)} \leq \sum_{t=1}^{T} m_i^{(t)} + \eta \sum_{t=1}^{T} \left| m_i^{(t)} \right| + \frac{2 \log n}{\eta}. \quad (11)$$

PROOF OF THEOREM 10. Suppose the algorithm did $T$ iterations without declaring infeasibility. Since the algorithm did not declare it infeasible, then we have that

$$\mathbf{p}^{(t)\top} A\mathbf{x}^{(t)} \geq \mathbf{p}^{(t)\top} \frac{b}{\alpha}$$

for every time step $t$. Thus, the left hand side of Eq. (11) is non-negative.

$$0 \leq \sum_{t=1}^{T} m_i^{(t)} + \eta \sum_{t=1}^{T} \left| m_i^{(t)} \right| + \frac{2 \log n}{\eta}$$

$$= \frac{1}{n} \sum_{t=1}^{T} (A_i \mathbf{x}^{(t)} - \frac{b_i}{\alpha}) + \eta T + \frac{2 \log n}{\eta}$$

Dividing by $T$, and choosing $\eta = \frac{\epsilon}{4n\alpha}$ and $T = \frac{32n^2\alpha^2 \log n}{\epsilon^2}$, we get

$$A_i \bar{\mathbf{x}} \geq \frac{b_i}{\alpha} - \eta n - \frac{2n \log n}{\eta T} \implies A_i \bar{\mathbf{x}} \geq \frac{b_i}{\alpha} - \frac{\epsilon}{2\alpha}.$$

The theorem statement then follows by choosing $\delta \leq \frac{1}{3}$ and with the observation that for some guess $B$ for the optimal value, we have $B \geq \frac{OPT}{1+\delta} \geq \frac{\epsilon}{1+\delta}$. □

## 3.3 Oracle for Cross-monotonic Sharing

We now consider the case where $h_{ij}(S)$ is cross-monotonic in $S$, and $u_i(S)$ is a non-decreasing submodular set function. Note that cross-monotonicity captures the Shapley value. We will present a $O(\log n)$ approximation to the oracle (Eq. (10)) for this setting, which when combined with Theorem 10, completes the proof of the first part

of Theorem 6. The key hurdle with devising an approximation algorithm is that the quantities $Q_{ij}$ in Eq. (10) can be negative; we show this is not an issue for cross-monotonic sharing.

*Simplifying the* DATA EXCHANGE *problem.* Before considering the oracle problem (Eq. (10)), we consider the overall DATA EXCHANGE problem (Eqs. (5) to (9)) and show some bounds for it. Let $u_{ij} := h_{ij}(\{j\}) = u_i(\{j\})$. Note that by cross-monotonicity, we have $h_{ij}(S) \leq u_{ij}$ for all $j \in S$.

LEMMA 12. *By losing a multiplicative factor of $(1 - \epsilon)$ in social welfare, for every $i$, we can set $x_{iS} = 0$ for any $S$ that contain some $j$ such that $u_{ij} := h_{ij}(\{j\}) \leq \frac{\epsilon^2}{n^2}$.*

PROOF. Fix some $i$. Let $S_{\text{small}} = \left\{ j \mid h_{ij}(\{j\}) \leq \frac{\epsilon^2}{n^2} \right\}$. Consider any solution $\mathbf{x}$. We claim that modifying $\mathbf{x}$ such that we add the value of $x_{iS}$ to $x_{iS \setminus S_{\text{small}}}$, and set $x_{iS} = 0$ only loses $(1 - \epsilon)$ factor in the objective. Since the utility sharing rule is cross-monotone, for any set $S$ we have $h_{ij}(S \setminus S_{\text{small}}) \geq h_{ij}(S)$ for all $j \in S \setminus S_{\text{small}}$. Further, we have $h_{ij}(S_{\text{small}}) \leq h_{ij}(\{j\})$ for all $j \in S_{\text{small}}$. Therefore, we have

$$u_i(S) = \sum_{j \in S} h_{ij}(S) = \sum_{j \in S_{\text{small}}} h_{ij}(S) + \sum_{j \in S \setminus S_{\text{small}}} h_{ij}(S)$$

$$\leq \sum_{j \in S_{\text{small}}} h_{ij}(\{j\}) + \sum_{j \in S \setminus S_{\text{small}}} h_{ij}(S \setminus S_{\text{small}})$$

$$\leq \frac{\epsilon^2}{n} + u_i(S \setminus S_{\text{small}}).$$

Adding up the losses, we lose a $\frac{\epsilon^2}{n}$ for each user $i$, leading to a loss of $\epsilon^2$ overall. By Lemma 9, the initial optimum was at least $\epsilon$. We therefore lose a factor of at most $(1 - \epsilon)$ in social welfare. □

We therefore assume $x_{iS} = 0$ for all $S$ s.t. $j \in S$ and $u_{ij} < \frac{\epsilon^2}{n^2}$.

*Approximating the Oracle.* For agent $i$, let

$$S^* = \arg\max \sum_{j \in S} Q_{ij} h_{ij}(S) \qquad OPT = \sum_{j \in S^*} Q_{ij} h_{ij}(S^*).$$

For given $\epsilon > 0$, the algorithm works as follows:

(1) Guess $OPT$ in powers of $(1 + \epsilon)$ by binary search.
(2) For constant $\delta = e - 1$, divide the agents into buckets based on the $Q_{ij}$ value. The $k^{th}$ bucket $B_k$ is defined as

$$B_k = \{j \mid Q_{ij} \in \left( u_0(1 + \delta)^k, u_0(1 + \delta)^{k+1} \right] \}$$

where $u_0 = \frac{\epsilon \cdot OPT}{n}$ and $k \in \{0, 1, \ldots, 3\lceil \log_{1+\delta}(\frac{n}{\epsilon}) \rceil - 1\}$.
(3) For each bucket $B_k$, let $V_k = \sum_{j \in B_k} Q_{ij} h_{ij}(B_k)$.
(4) For this guess of $OPT$, the final solution is $S_z$ where $z = \arg\max_k V_k$.
(5) The solution is valid for this value of $OPT$ if $V_z \geq OPT/\hat{\alpha}$, where $\hat{\alpha} = 3e(1 + 3\epsilon) \ln n$. We use the largest $OPT$ for which the solution returned is valid, and return this solution.

In the analysis below, we assume $OPT$ can be precisely guessed.

THEOREM 13. *For $\epsilon > 0$, when utilities $u_i(S)$ are monotone non-decreasing in $S$ and the utility sharing rule is cross-monotone, the* ORACLE *problem can be approximated to factor $\alpha \leq 3e(1 + 2\epsilon) \ln n$ in $O(\frac{n \log n}{\log(1+\epsilon)})$ time and correspondingly many calls to* MLSUB.

Proof. Let $S_0 = \{j \in S^* | Q_{ij} < 0\}$. We have:

$$\sum_{j \in S^*} Q_{ij} h_{ij}(S^*) = \sum_{j \in S_0} Q_{ij} h_{ij}(S^*) + \sum_{j \in S^* \setminus S_0} Q_{ij} h_{ij}(S^*)$$

$$\leq \sum_{j \in S^* \setminus S_0} Q_{ij} h_{ij}(S^*) \leq \sum_{j \in S^* \setminus S_0} Q_{ij} h_{ij}(S^* \setminus S_0).$$

where the final inequality follows by cross-monotonicity. Since $S^*$ is optimal, this means $S_0 = \emptyset$. Therefore, we assume $Q_{ij} > 0$.

Next note that $OPT \geq \sum_{j \in S} Q_{ij} h_{ij}(S)$ for $S = \{j\}$, which means $Q_{ij} u_{ij} \leq OPT$ for all $j$. Given constant $\epsilon \in (0, 1]$, let $S_{\text{small}} = \left\{ j \mid Q_{ij} u_{ij} < \epsilon \cdot \frac{OPT}{n} \right\}$. By the same argument as in the proof of Lemma 12, we can restrict to agents in $X \setminus S_{\text{small}}$ by losing a $(1 - \epsilon)$ factor in $OPT$. Let $\hat{X} = X \setminus S_{\text{small}}$, so that these are now the only agents of interest. The above implies $Q_{ij} u_{ij} \in OPT \cdot \left[ \frac{\epsilon}{n}, 1 \right]$ for $j \in \hat{X}$. Since $u_{ij} \in \left[ \frac{\epsilon^2}{n^2}, 1 \right]$ by Lemma 12, this implies $Q_{ij} \in OPT \cdot \left[ \frac{\epsilon}{n}, \frac{n^2}{\epsilon^2} \right]$. Therefore, the buckets constructed by the algorithm only use agents from $\hat{X}$.

Let $\hat{S} = S^* \cap \hat{X}$. By the Pigeonhole principle, the elements of some bucket must contribute at least $\frac{\log(1+\delta)}{3 \log \frac{n}{\epsilon}}$ fraction of the objective, $OPT$. Suppose this is the $k^{th}$ bucket $B_k$. We therefore have:

$$\frac{\log(1 + \delta)}{\log \frac{n}{\epsilon}} \cdot OPT \leq \sum_{j \in \hat{S} \cap B_k} Q_{ij} h_{ij}(\hat{S}) \leq (1 + \delta)^{k+1} \sum_{j \in \hat{S} \cap B_k} h_{ij}(\hat{S}).$$

Suppose we choose $B_k$ as the solution instead. We have

$$\sum_{j \in B_k} Q_{ij} h_{ij}(B_k) \geq (1 + \delta)^k \sum_{j \in B_k} h_{ij}(B_k) = (1 + \delta)^k u_i(B_k)$$

$$\geq (1 + \delta)^k u_i(B_k \cap \hat{S}) = (1 + \delta)^k \sum_{j \in B_k \cap \hat{S}} h_{ij}(B_k \cap \hat{S})$$

$$\geq (1 + \delta)^k \sum_{j \in B_k \cap \hat{S}} h_{ij}(\hat{S}) \geq \frac{(1 - \epsilon) \log(1 + \delta)}{3(1 + \delta) \log \frac{n}{\epsilon}} OPT.$$

Here, the second inequality holds because $u_i$ is monotonically non-decreasing, and the next inequality holds since $h_{ij}$ is cross-monotone, so that $h_{ij}(B_k \cap \hat{S}) \geq h_{ij}(\hat{S})$. Thus, the largest of the solutions $V_k$ is a $\frac{3(1+\delta) \log \frac{n}{\epsilon}}{(1-\epsilon) \log(1+\delta)}$-approximation to the optimal solution. This is minimized at $\delta = e - 1$, giving us an approximation ratio of $\frac{3e \log \frac{n}{\epsilon}}{(1-\epsilon)} \leq 3e(1 + 2\epsilon) \log n$ for $\epsilon < \frac{1}{2}$ and for large enough $n$.

We can execute this algorithm in almost linear time in the following way: guess the right value of $OPT$ by a binary search, which takes $O(\frac{\log n}{\log(1+\epsilon)})$ time to find $OPT$ up to multiplicative error of $(1 + \epsilon)$. Throw out all elements that have $Q_{ij} u_{ij} < \frac{\epsilon \cdot OPT}{n}$ and $u_{ij} < \frac{\epsilon^2}{n^2}$, and find the bucket with the largest utility. This takes time $O(n)$, leading to an overall time of $O(\frac{n \log n}{\log(1+\epsilon)})$. □

## 3.4 Oracle for Symmetric Weighted Utilities

We will now complete the proof of the second part of Theorem 6. Recall that in the symmetric weighted setting, each agent $j$ has a fixed non-negative amount of data $s_{ij}$ that they contribute agent $i$ if they trade. For $S \subseteq X \setminus \{i\}$, let $D(S) = \sum_{j \in S} s_{ij}$. Then the utility that $i$ receives if it is assigned $S$ is $f_i(D(S))$, where $f_i$ is a monotonically non-decreasing and non-negative concave function. Further, we divide the utilities according to the proportional value rule $h_{ij}(S) = \frac{f_i(D(S))}{D(S)} s_{ij}$.

Let $Q(S) = \sum_{j \in S} Q_{ij} s_{ij}$. Then the oracle (Eq. (10)) becomes:

$$\max_{S \subseteq X \setminus \{i\}} \frac{f_i(D(S))}{D(S)} \cdot Q(S).$$

We will present a $(1+\epsilon)$ approximation to the oracle, which when combined with Theorem 10 completes the proof of the second part of Theorem 6. First note that $f_i(x)/x$ is a non-increasing function of $x$. Therefore, the optimal solution does not contain any $j$ with $Q_{ij} \leq 0$; if it did, removing this $j$ increases $Q(S)$ and does not decrease $\frac{f_i(D(S))}{D(S)}$. We therefore assume $Q_{ij} > 0 \; \forall j$.

For $\epsilon > 0$, the algorithm is now as follows, where $S^*$ is the optimal solution:

(1) Guess the value of $D(S^*)$ in powers of $(1 + \epsilon)$. Let $\phi$ denote the current guess.

(2) Solve the following KNAPSACK problem to a $(1 + \epsilon)$ approximation [35]:

$$V(\phi) = \max_S \sum_{j \in S} Q_{ij} s_{ij} \qquad \text{s.t.} \qquad \sum_{j \in S} s_{ij} \leq \phi.$$

(3) Choose $\phi$ and the solution that maximizes $V(\phi) \cdot \frac{f_i(\phi)}{\phi}$.

Theorem 14. *The above algorithm yields a $(1 + \epsilon)$ approximation to the oracle in polynomial time.*

Proof. Let $\phi^* = \sum_{j \in S^*} s_{ij}$, and let $V(\phi^*) = \sum_{j \in S^*} Q_{ij} s_{ij}$. Our algorithm tries some $\hat{\phi} \in \phi^* \cdot [1, 1 + \epsilon]$. Fix this choice of $\hat{\phi}$. Clearly,

$$\frac{f_i(\phi^*)}{\phi^*} \leq (1 + \epsilon) \frac{f_i(\hat{\phi})}{\hat{\phi}} \qquad \text{and} \qquad V(\phi^*) \leq V(\hat{\phi}).$$

Combining these yields the proof. □

## 4 EXPERIMENTS

We will now empirically compare the performance of our approximation algorithm in Section 3 with a no-sharing baseline, and with a pair-wise trade benchmark, showing we outperform both. In our experiments, each agent corresponds to a path in a road network. The delay of each edge in the road network is a random variable and each agent has a set of samples for each edge on its path that it can trade with other agents. The goal of each agent is to trade her samples in order minimize the sample variance in the estimate of the delay on her path.

*Setup.* We sample a random neighborhood of radius 8 from the Manhattan road network in [1]. This will serve as the graph of interest for the rest of the experiment. We have $n = 20$ agents. Each agent $i$ is assigned a path in the graph in the following way: Sample a random node $u$ in the graph. Sample a length $t$ uniformly at random between 5 and the depth of the BFS tree from $u$. Sample a node $v$ uniformly at random at layer $t$ of the BFS tree. The shortest path from $u$ to $v$ in the graph is the path $P_i$ corresponding to agent $i$, and she is interested in minimizing the variance of the sample mean of the delay of this path.

The delay of each edge $e$ is a random variable whose variance $\sigma_e^2$ is drawn uniformly from $[0, 1]$, independently of other edges. Agent

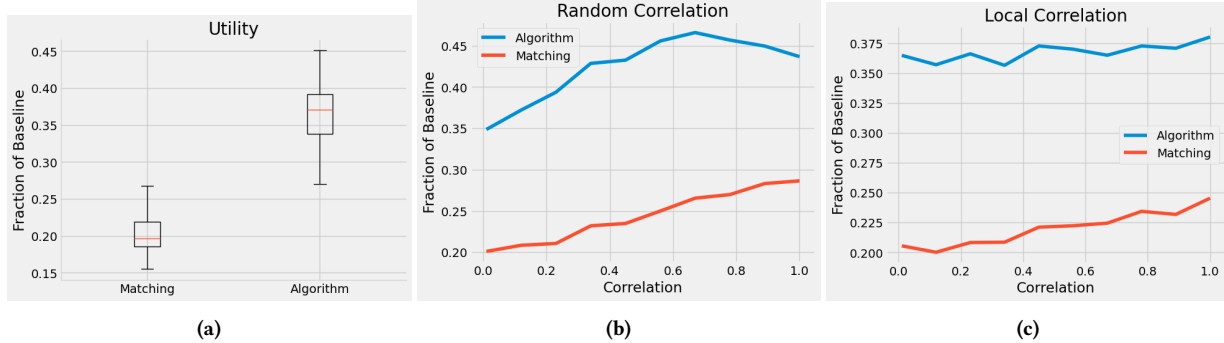

**Figure 1: (a) Box plots of the total utility of the algorithm and benchmark (matching) solutions, measured as a fraction of the baseline. (b,c) Total utility of the algorithm and matching benchmark with varying levels of correlation, again measured as a fraction of the baseline. Figure (b) is Random correlation, and (c) is Local correlation.**

$i$ starts with $z^{(i)}$ data points for the delay of her path $P_i$, where $z^{(i)}$ is chosen uniformly at random between 2 and 9. Therefore, she starts with $z_e^{(i)} = z^{(i)}$ data points for each edge $e$ in her path.

The agent's objective is to minimize the sum of the sample variances of the delays of the edges in her path $P_i$. Her initial sample variance is $\frac{\sigma_e^2}{z_e^{(i)}}$ and therefore, her initial total sample variance is

$$\text{Baseline for } i = v_0(i) := \sum_{e \in P_i} \frac{\sigma_e^2}{z_e^{(i)}}$$

Suppose she receives data from a set of other agents $S$, who collectively give her $z_e^{(S)}$ additional samples for edge $e$. Then, her utility is defined as the reduction in total sample variance. That is,

$$\text{Utility of } i = u_i(S) = v_0(i) - \sum_{e \in P_i} \frac{\sigma_e^2}{z_e^{(i)} + z_e^{(S)}}.$$

This is a monotonically increasing submodular function. We perform the cost-sharing via the Shapley value. We simulate the Shapley value by taking $m = 10$ random permutations, and use use $\epsilon = 0.01$ as the violation allowed in the BALANCE constraints.

*Results.* Since the optimal solution to DATA EXCHANGE is NP-Hard, we compare the total utility of our approximation algorithm (Section 3.3) to the baseline sample variance $\sum_i u_0(i)$, where no agents in the solution share their data. As a benchmark, we also find the best solution with trades only between pairs of agents. For this, we construct a graph on the agents where the weight for pair $(i, j)$ is the maximum utility of DATA EXCHANGE with $\epsilon$-BALANCE on just these two agents. We then find a maximum weight matching on this weighted graph. (See Appendix D.2 for more details.)

In Fig. 1a, we present the total utility of our algorithm and the matching benchmark, measured as a fraction of the baseline sample variance, across several random samples of the road network. Our algorithm outperforms the benchmark, by a factor of 1.8 on average. Note that we can easily construct instances with a single long path with $m$ edges and many paths sharing one edge with this path, where our algorithm outperforms matching by a factor of $\Omega(m)$. The goal of our experiment is to show that our algorithm has a significant advantage even in more realistic settings.

We now introduce *correlation* between the random variables of the edges. In this setting, we assume that correlated edges have their delays sampled from the same distribution. We introduce this correlation in two ways. In *random* correlation (Fig. 1b), we sample pairs of edges uniformly at random and correlate the pair. We measure the correlation ($x$-axis) as a ratio of the number of pairs sampled to the total number of edges in the graph. In *local* correlation (Fig. 1c), we sample vertices uniformly at random, and correlate all the edges incident to this edge. We measure the correlation ($x$-axis) as a ratio of the number of vertices sampled to the total number of vertices in the graph.

We measure how the total utility of our algorithm and the matching benchmark changes as a function of the correlation in Figs. 1b and 1c, again measured as a fraction of the baseline sample variance. Our algorithm outperforms the benchmark in both modes of correlation, and at both high and low levels of correlation.

## 5   CONCLUSION

There are several open questions that arise from our work. First, the approximation ratio for Shapley value sharing is $O(\log n)$ and we have not ruled out the existence of a constant approximation. Secondly, our algorithmic results require utilities to be submodular. Though this is a natural restriction, there are cases where it does not hold. For instance, if each dataset is a collection of features, the effect of combining features could be super-additive [17]. Devising efficient algorithms for special types of non-submodular functions that arise in learning is an interesting open question.

Next, for Shapley value sharing (as opposed to proportional sharing), our negative result for core-stability (Theorem 20) only shows the absence of a $(2 - \epsilon)$-approximation to welfare. Either strengthening this impossibility result or showing a constant approximation that lies in the exact core would be an interesting question. Further, it would be interesting to study strategyproofness for thick or random markets, analogous to results for stable matchings [7, 22].

Finally, our model can be viewed as budget balance with a single global price per unit utility transferred. Though there are hurdles to defining an Arrow-Debreu type market with endogenous prices for each data type, it would be interesting to define a richer and tractable class of markets along this direction.

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

## A COMPARISON TO KIDNEY EXCHANGE

It is instructive to compare the upper and lower bounds for social welfare in Section 2 with those for barter with non-replicable goods, that is, kidney exchange [2, 30]. Note that there, trades are deterministic and hence intractability with long sequences of exchanges follows from the hardness of set-packing type problems. On the other hand, our formulation of Data Exchange Problem allows interim balance and its intractability (Theorem 5) is because of non-linear utility functions, making it technically very different. Similarly, the positive approximation results in Theorem 6 are very different from the $\frac{k+1}{3}$ approximation factor for length $k$-trades in kidney exchange, that follow from approximation algorithms for maximum set packing [13, 20, 34].

Similarly, for kidney exchange, stability and strategyproofness can be simultaneously achieved with Pareto-efficiency or maximizing match size; see [6, 21, 30, 32] for positive results in various exchange models. However, for Data Exchange Problem, we show in Appendix D that these goals are incompatible with welfare, mainly because of the non-linearity in utility functions. Nevertheless, the Greedy matching rules from kidney exchange does extend to our setting and is simultaneously strategyproof and 2-stable.

## B NP-HARDNESS OF WELFARE MAXIMIZATION: PROOF OF THEOREM 5

We now show that welfare maximization is NP-Hard for submodular utilities and Shapley value sharing. We note that the hardness result holds even when for any $S$, both $u_i(S)$ and $h_{ij}(S)$ are computable in near-linear time.

Proof of Theorem 5. We reduce from an instance of Exact Cover by 3-Sets (X3C). In this problem there are $m$ sets $P_1, \ldots, P_m$, each containing three elements. The universe $U$ has $3k$ elements $\{e'_1, e'_2, \ldots, e'_{3k}\}$ and the decision problem is whether there are $k$ disjoint sets from $P_1, \ldots, P_m$ that exactly cover $U$.

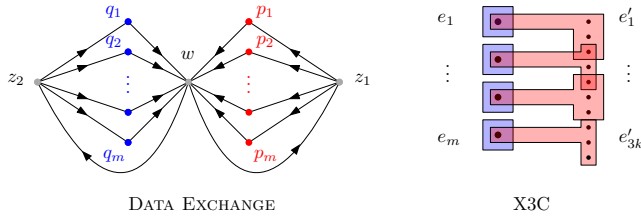

**Figure 2: Construction for Theorem 5. The X3C instance has elements labelled. Blue boxes correspond to $Q_i$s and red boxes correspond to $P_i$s.**

We reduce from the X3C instance in the following way: Add $m$ dummy elements $e_1, \ldots, e_m$ to $U$. Modify each set $P_i$ so that it also includes the dummy element $e_i$. We also add $m$ new sets $Q_1, \ldots, Q_m$ where $Q_i = \{e_i\}$.

In the corresponding DATA EXCHANGE instance, we have two sets of agents: $\{p_i\}_{i \in [m]}$ and $\{q_i\}_{i \in [m]}$ and three special agents $w, z_1, z_2$. For each $i \in [m]$, we add the directed edges $(p_i, w)$, $(q_i, w)$, $(z_1, p_i)$,and $(z_2, q_i)$. We also have the edges $(w, z_1)$ and $(w, z_2)$. Agents can only send data along a directed edge. See Fig. 2.

We have a one-to-one correspondence between the agents $p_i$ and the sets $P_i$, and the agents $q_i$ and the sets $Q_i$. Let $V = \{p_i\}_{i \in [m]} \cup \{q_i\}_{i \in [m]}$. For agent $a \in V$, let elem($a$) denote the set of elements in $U$ covered by the set corresponding to $a$. For $S \subseteq V$, let sets($S$) $\subseteq \{P_1, \ldots, P_m, Q_1, \ldots, Q_m\}$ denote the the corresponding sets in the X3C instance. Then the utility of agent $w$ is defined as[3]

$$u_w(S) = \text{Number of elements from } U \text{ covered by sets}(S).$$

The Shapley value implies the following: Given $S \subseteq V$, let element $e \in U$ be covered by $c_e(S)$ sets from sets($S$). Then each agent $a \in S$ whose corresponding set covers $e$ contributes utility $\frac{1}{c_e(S)}$. Then the utility share $h_{wa}(S)$ for $a \in S$ is

$$h_{wa}(S) = \sum_{e \in \text{elem}(a)} \frac{1}{c_e(S)}$$

For every $i \in [m]$, we set $u_{p_i}(\{z_1\}) = 4, u_{q_i}(\{z_2\}) = 1$. We also set $u_{z_1}(\{w\}) = \frac{7k}{2}$ and $u_{z_2}(\{w\}) = m - \frac{k}{2}$. The Shapley values are trivial to define.

The decision problem is whether there exists a feasible DATA EXCHANGE solution with social welfare exactly $3(m + 3k)$. We show that if the X3C instance is a **YES** instance, the welfare is exactly $3(m + 3k)$, while for **NO** instances, the welfare is strictly smaller.

*Case 1.* If the X3C instance is a **YES** instance, this means there are sets (w.l.o.g.) $P_1, \ldots, P_k$ that cover all elements from $\{e'_1, \ldots, e'_{3k}\}$. For the set $T = \{p_1, \ldots, p_k, q_1, \ldots, q_m\}$, set $x_{wT} = 1$. Set $x_{p_i\{z_1\}} = \frac{7}{8}$ for all $i \in [k]$. Set $x_{q_i\{z_2\}} = \frac{1}{2}$ for $i \in [k]$ and $x_{q_i\{z_2\}} = 1$ for $i \notin [k]$. Further, set $x_{z_1\{w\}} = 1$ and $x_{z_2\{w\}} = 1$. All other values of $x$ are set to zero. It can be checked that this is a feasible solution with total utility $3(m + 3k)$.

[3]These utilities are not bounded above by 1, but we can achieve this by simply scaling them down by the largest possible utility. This does not change the reduction.

*Case 2.* Suppose we have a **NO** instance of X3C. This means any collection of $k$ sets covers less than $3k$ elements from $\{e'_1, \ldots, e'_{3k}\}$. Observe that $u_{z_1}(\{w\}) + u_{z_2}(\{w\}) = m + 3k$, so that by the BALANCE condition on feasibility, we have

$$\sum_S x_{wS} u_w(S) \leq m + 3k.$$

Also observe that from the balance constraint, the social welfare is at most thrice the utility that $w$ gets. Therefore, to achieve a total utility of $3(m + 3k)$, it must be that $w$ receives a utility of $m + 3k$ and provides a utility of $\frac{7k}{2}$ to $z_1$ and $m - \frac{k}{2}$ to $z_2$. Since any set $T$ provides $w$ with utility at most $m + 3k$ (as there are $m + 3k$ elements in $U$), the previous statement implies that if $x_{wT} > 0$ then $u_w(T) = m + 3k$. Thus, each such $T$ must correspond to sets $S$ that cover all the elements in $\{e'_1, \ldots, e'_{3k}\}$. Since the original X3C instance was a **NO** instance, this means that each such $T$ must contain at least $k + 1$ agents from $\{p_i\}_{i \in [m]}$. But note that in any such $T$, the agents $q_1, \ldots, q_m$ can collectively only get a utility of at most $m - \frac{k+1}{2}$, since each element $e_j$ covered by $Q_j$ and $P_j$, where $p_j \in T$ contributes utility $1/2$ to $Q_j$. This bounds the expected utility that users in $q_1, \ldots, q_m$ receive by $m - \frac{k+1}{2}$. This then means that $z_2$ can collectively give (and hence receive) a utility of at most $m - \frac{k+1}{2}$, which is a contradiction. Therefore the total utility of the solution is strictly smaller than $3(m + 3k)$, completing the reduction. □

## C   GENERAL CONCAVE UTILITIES WITH PROPORTIONAL SHARING

In this section, we will prove Theorem 7. Unlike Section 3, we will assume the utilities are continuous and concave, which is motivated by agents partially sharing their data. Formally, we assume agent $j$ can contribute at most $s_{ij}$ amount of data to $i$. Suppose they contribute fraction $y_{ij} \in [0, 1]$ of this data. Denote by $\vec{y_i}$ the vector $\langle y_{i1}, y_{i2}, \ldots \rangle$. Then the utility $i$ receives is given by the monotonically non-decreasing concave function $u_i(\vec{y_i}) \in [0, 1]$.

We will assume $u_i(\vec{0}) = 0$. Further, we will assume $i$ gets strictly positive utility from every agent $j$'s data[4], so that for a value $\delta > 0$ with polynomial bit complexity, we assume $u_i(\vec{y_i}) \geq \delta$ for any $\vec{y_i}$ with at least one coordinate set to 1.

We now assume this utility is shared via proportional value as:

$$h_{ij}(\vec{y_i}) = \frac{s_{ij}y_{ij}}{\sum_{k \in X \setminus \{i\}} s_{ik}y_{ik}} \cdot u_i(\vec{y_i}).$$

Let $Y_i = \prod_{j \in X \setminus \{i\}} [0, 1]$. Then a solution to DATA EXCHANGE assigns a Borel measure $\mu_i$ to $Y_i$, and it satisfies the BALANCE conditions:

$$\forall i, \sum_{j \in X \setminus \{i\}} \int h_{ij}(\vec{y_i}) \, d\mu_i = \sum_{j \in X \setminus \{i\}} \int h_{ji}(\vec{y_j}) \, d\mu_j. \quad (12)$$

The objective of maximizing social welfare becomes:

$$\text{Social Welfare} = \sum_{i \in X} \sum_{j \in X \setminus \{i\}} \int h_{ij}(\vec{y_i}) \, d\mu_i.$$

[4]We assume there is a set $X_i$ of agents satisfying this condition and these are the only agents of interest in the remaining optimization. For simplicity, we are assuming $X_i = X$.

## C.1 Multiplicative Weight Method and Oracle

As before, we can solve this using the multiplicative weight method, where the set $P$ is simply the set of all Borel measures on $Y_i$ for each agent $i$. This solution will violate the balance constraint by an additive $\epsilon$. For a given choice of dual variables $\{Q_{ij}\}$, the optimum Borel measure for the oracle becomes a point mass for each agent $i$. For given $i$, omitting the calculation, the oracle corresponds to solving the problem:

$$\text{Oracle for agent } i = \max_{\vec{y}_i \in Y_i} \left( \sum_{j \in X\setminus\{i\}} Q_{ij} s_{ij} y_{ij} \right) \cdot \left( \frac{u_i(\vec{y}_i)}{\sum_{j \in X\setminus\{i\}} s_{ij} y_{ij}} \right).$$

For this agent $i$ and setting of dual variables, let $OPT$ denote the optimal value of the oracle. We now show how to approximate $OPT$ to a factor of $1 + \epsilon$ in polynomial time.

*Approximating the Oracle.* First note that we cannot simply delete agents $j$ with $Q_{ij} < 0$; the optimum solution can include such agents. However, if all agents have $Q_{ij} < 0$, then $OPT = 0$, since $\vec{y}_i = \vec{0}$ is a feasible solution. We can easily check this; therefore, we assume $OPT > 0$. In this case, there exists agent $j$ such that $y_j = 1$ in the optimal solution. Otherwise, we can scale the variables up till this is satisfied; note that the ratio of the two linear terms remains unaffected by scaling, while $u_i(\vec{y}_i)$ only increases by scaling up.

Let $V(\vec{y}_i) = \frac{u_i(\vec{y}_i)}{\sum_{j \in X\setminus\{i\}} s_{ij} y_{ij}}$, and let $V_{\min}$ and $V_{\max}$ denote the maximum and minimum values attained by this function over $\vec{y}_i \in Y_i$ with at least one coordinate set to 1. Note that both these quantities have polynomial bit complexity since $u_i(\vec{y}_i) \geq \delta$ for such solutions $\vec{y}_i$. Let $V^*$ denote its value in the optimal solution, $OPT$; clearly $V^* \in [V_{\min}, V_{\max}]$.

Our algorithm has the following steps:

(1) We use binary search to guess $V^*$ to a factor of $1 + \epsilon$ in the range $[V_{\min}, V_{\max}]$.

(2) For each guess $V$, we solve the following convex optimization problem:

$$\max_{\vec{y}_i \in Y_i} \sum_{j \in X\setminus\{i\}} Q_{ij} s_{ij} y_{ij}, \qquad \text{s.t.} \qquad u_i(\vec{y}_i) \geq V \cdot \sum_{j \in X\setminus\{i\}} s_{ij} y_{ij}.$$

(3) Among these solutions $\vec{y}_i$, one for each guess of $V$, we choose the one that has largest value for the oracle objective.

LEMMA 15. *For any $\epsilon > 0$ above algorithm is a $1 + \epsilon$ approximation to $OPT$ in time polynomial in the input bit complexity and $\frac{1}{\epsilon}$.*

PROOF. The algorithm tries some $V \in [V^*/(1+\epsilon), V^*]$. For this setting, suppose the convex program achieves objective $W$. Since $OPT$'s variables $\vec{y}_i^*$ are feasible for this convex program, we have $W \geq \sum_{j \in X\setminus\{i\}} Q_{ij} s_{ij} y_{ij}^*$. This means the oracle objective achieved by our algorithm satisfies:

$$\text{Oracle Objective} \geq W \cdot V \geq \sum_{j \in X\setminus\{i\}} Q_{ij} s_{ij} y_{ij}^* \cdot \frac{V^*}{1+\epsilon},$$

completing the proof. $\square$

This finally shows the following theorem, which restates Theorem 7:

THEOREM 16. *For general monotone concave utilities with proportional value sharing, for any $\epsilon > 0$, there is an algorithm for DATA EXCHANGE running in time polynomial in input size and $\frac{1}{\epsilon}$, that approximates the social welfare to a factor of $(1 + \epsilon)$ while violating Eq. (12) by an additive $\epsilon$.*

# D CORE STABILITY AND STRATEGYPROOFNESS

We will now consider the concept of core stability as defined in Definition 8. We assume utilities are monotone set functions, as in Section 3. We first show that such solutions always exist to arbitrarily good approximations, and the special case of 2-stable solutions can be efficiently computed via a GREEDY matching algorithm. We then show that core solutions can be far from welfare optimal; nevertheless, approximate core stability trades off with approximate welfare.

We finally consider strategic misreports by agents in Appendix D.4, where we present a formal model and show how strategyproofness trades off with approximate welfare maximization.

## D.1 Existence of Core

We now show the existence of an $\epsilon$-approximate core solution (Definition 8) for any $\epsilon > 0$. This solution is defined as follows, where we note in the definition below that $\mathcal{F}'$ is a feasible solution constructed just on agents in $U$.

DEFINITION 17. *For $\epsilon > 0$, a feasible solution $\mathcal{F}$ to DATA EXCHANGE is $\epsilon$-approximately core stable if there is no $U \subseteq X$ of users and another feasible solution $\mathcal{F}'$ just on the users in $U$ such that for all $i \in U$, $u_i(\mathcal{F}') > u_i(\mathcal{F}) + \epsilon$.*

Our proof uses Scarf's lemma from cooperative game theory [31]. More recently, this lemma has seen applications in showing existence of stable matchings [26]. However, Scarf's lemma is an existence result, and it is not clear how such a solution can be efficiently computed.

THEOREM 18. *For the DATA EXCHANGE problem with utilities being non-negative monotone set functions, an $\epsilon$-approximately core-stable solution always exists for any fixed $\epsilon > 0$.*

PROOF. Define a matrix $Q$ as follows, where the rows are agents and there is a column for every coalition $S$ and every possible feasible solution $f$ to DATA EXCHANGE just on the agents in $S$. We denote this feasible solution by $f$, and let $set(f) = S$.

To make the number of columns finite, we round utilities down to the nearest $\epsilon$ to create the set $\hat{U} = \{0, \epsilon/2, \epsilon, \dots, 1\}$. The columns of $Q$ are the elements of the set $\hat{U}^n$. Given a solution $f$, let $u^f = (u_1(f), u_2(f), \dots, u_n(f))$ denote the utilities of the $n$ agents. We round each $u_i(f)$ down to the nearest multiple of $\epsilon/2$ yielding utility $\hat{u}_i(f)$. The resulting vector of utilities is an element of $\hat{U}^n$, which we denote as $col(f)$. Let $g = col(f)$.

Let $Q_{ig} = 1$ iff $\hat{u}_i(g) > 0$. Then there is a natural ordering $\succeq_i$ of the columns $g$ with $Q_{ig} = 1$, such that $g^1 \succeq_i g^2 \iff \hat{u}_i(g^1) \geq \hat{u}_i(g^2)$.

Consider choosing $g$ to fraction $y_g$, so that for every agent $i$, we have $\sum_g Q_{ig} y_g \leq 1$. Consider any feasible solution $f^*$ to DATA EXCHANGE. Since $g^* = col(f^*)$ is a column of $f^*$, choosing $y_{g^*} = 1$,

this shows the set of feasible solutions is captured by the constraints $T = \{\sum_g Q_{ig} y_g \leq 1 \forall i\}$. Similarly, any solution to the constraints $T$ maps to at least one feasible solution to DATA EXCHANGE. To see this, set

$$\hat{x}_{iS} = \sum_{f:i \in \text{set}(f)} y_f \cdot x_{iS}^f,$$

where $\{x_{iS}^f\}$ are the variables corresponding to any solution $f$ such that $g = \text{col}(f)$. The variables $\{\hat{x}_{iS}\}$ preserve Eqs. (1) and (2), showing feasibility.

Scarf's lemma [31] (see Lemma 3.1 in [26]) then says that there is a $y$ satisfying constraints $T$, such that for every column $g$ of $Q$, there is a row $i$ with $Q_{ig} > 0$ (i.e. an agent) such that both these conditions hold:

(1) $\sum_{g'} Q_{ig'} y_{g'} = 1$; and
(2) For every $g'$ with $Q_{ig'} = 1$ and $y_{g'} > 0$, we have $f' \succeq_i f$, that is $\hat{u}_i(g') \geq \hat{u}_i(g)$.

Taking the linear combination of the second condition using weights given by $y$, we have

$$\sum_{g' | Q_{ig'} = 1} y_{g'} \cdot \hat{u}_i(g') \geq \hat{u}_i(g).$$

Consider some feasible solution to DATA MARKETS given by $y$. Then, the above says that for every possible $f$ that a set of agents $\text{set}(f)$ could deviate to, there is one agent $i$ whose utility (as given by $\hat{u}$) in $y$ (the LHS of the previous equation) is at least $\hat{u}_i(f)$. Since we discretized utilities to within $\epsilon/2$, this means this coalition will not deviate to $f$ if they are indifferent to utilities increasing by $\epsilon$. This shows $y$ yields an $\epsilon$-approximate core solution, completing the proof. □

## D.2 Greedy Matching is 2-Stable

The above proof is based on a fixed point argument, and does not lend itself to efficient computation. On the positive side, we show a simple polynomial time algorithm for the special case of 2-stability. Recall from Definition 8 that a solution is $c$-stable if there is no coalition of at most $c$ agents that can deviate to improve all their utilities. The algorithm below generalizes a similar greedy algorithm for kidney exchanges [30].

GREEDY *Algorithm.* Consider a pair of agents $(i, j)$. Let $u_{ij} = u_i(\{j\})$ and $u_{ji} = u_j(\{i\})$. Consider the DATA EXCHANGE solution that sets $x_{ij} = \min\left(1, \frac{u_{ji}}{u_{ij}}\right)$ and $x_{ji} = \min\left(1, \frac{u_{ij}}{u_{ji}}\right)$. This solution satisfies BALANCE for this pair of agents, and is maximal, in the sense that both agents cannot simultaneously improve their utilities. Both agents achieve utility $\hat{u}_{ij} = \min(u_{ij}, u_{ji})$ in this solution.

Now construct a graph on the agents where we place an edge between every pair of agents $i$ and $j$ with weight $w_{ij} = \hat{u}_{ij}$. Find any greedy maximal weight matching in this graph (call this algorithm GREEDY), and for each pair of agents in this matching, construct the DATA EXCHANGE solution for this pair. This yields the final solution GREEDY.

THEOREM 19. *For arbitrary monotone utility functions of the agents, the* GREEDY *algorithm is 2-stable.*

PROOF. Suppose not, then there is a pair $(i, j)$ that can deviate and where $\hat{u}_{ij}$ is larger than the utilities $i$ and $j$ were receiving

in GREEDY. But by the construction of maximal weight matching, one of $i$ or $j$ must have larger utility in the matching, which is a contradiction. Therefore, GREEDY is 2-stable. □

## D.3 Gap between Social Welfare and the Core

We next study the tradeoff between the core and social welfare. We first show that though core-stable solutions always exist, they may be quite far from welfare optimal solutions.

THEOREM 20. *For symmetric weighted utilities and proportional sharing of utility, the gap in social welfare between any core-stable solution and the welfare-optimal solution can be $\Omega(\sqrt{n})$.*

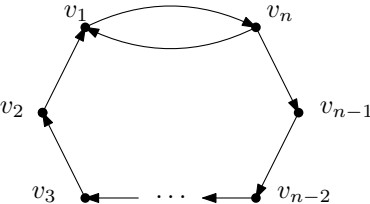

**Figure 3: Instance for the proof of Theorem 20**

PROOF. Recall the definition of symmetric weighted utilities from Section 3.4. Now consider the instance in Fig. 3. To simplify notation, we denote by $s_{ij}$ the weight along the directed edge $(v_j, v_i)$, the utility function at $v_i$ as $u_i$, etc. Note that $s_{ij} = 0$ if there is no directed edge $(v_j, v_i)$. We set $s_{1n} = s_{n1} = M$, and $s_{ij} = 1$ for the remaining directed edges $(v_j, v_i)$, where $M \geq 3$ will be chosen later. We also set $f_i(x) = \sqrt{x}$ for all $i$. Note that

$$u_1(\{2, n\}) = \sqrt{M+1}; \quad u_1(\{2\}) = 1; \quad u_1(\{n\}) = u_n(\{1\}) = \sqrt{M},$$

and further, $h_{12}(\{2, n\}) = \frac{1}{\sqrt{M+1}}$ and

$$h_{12}(\{2\}) = h_{23}(\{3\}) = \cdots = h_{n-1n}(\{n\}) = 1.$$

To lower bound social welfare, consider the solution that sets $x_1(\{2\}) = x_2(\{3\}) = \cdots = x_{n-1}(\{n\}) = 1$ and sets $x_n(\{1\}) = \frac{1}{\sqrt{M}}$. This solution has welfare $n$.

We now upper bound the welfare of any core solution. In this solution, denote $p = x_1(\{2\})$, $q = x_1(\{2, n\})$, and $r = x_1(\{n\})$. Denote the utility of agent $i$ in this solution as $U_i$. We have

$$U_2 = p \cdot h_{12}(\{2\}) + q \cdot h_{12}(\{2, n\}) = p + q \cdot \frac{1}{\sqrt{M+1}} \leq p + \frac{1}{\sqrt{M+1}}. \tag{13}$$

Note that $U_1, U_n \leq \sqrt{M+1}$. By balance, all of $\{2, 3, \ldots, n-1\}$ have the same utility. Therefore, the total utility of the solution is at most

$$\text{Total Utility} \leq (n-2) \cdot U_2 + 2\sqrt{M+1}$$

By balance, we have $U_1 = U_n$. Further, if $U_1 < \sqrt{M}$, then agents $\{1, n\}$ can deviate and obtain utility $\sqrt{M}$ each by setting $x_1(\{n\}) = x_n(\{1\}) = 1$. Since this is not possible, we have $U_1 \geq \sqrt{M}$. Therefore,

$$U_1 = p + q \cdot \sqrt{M+1} + r \cdot \sqrt{M} \geq \sqrt{M}.$$

Since $p + q + r \leq 1$, the above implies

$$p+(1-p)\cdot\sqrt{M+1} \geq \sqrt{M} \quad \Rightarrow \quad p \leq \frac{\sqrt{M+1}-\sqrt{M}}{\sqrt{M+1}-1} \leq \frac{1}{\sqrt{M+1}},$$

where the final inequality holds for $M \geq 3$. Plugging this into Eq. (13), we have $U_2 \leq \frac{2}{\sqrt{M+1}}$, so that

$$\text{Total Utility} \leq (n-2)\cdot\frac{2}{\sqrt{M+1}} + 2\sqrt{M+1} \leq 4\sqrt{n-2},$$

where we set $M = n - 3$. This shows a gap of $\Omega(\sqrt{n})$ between the welfare of the core and the social optimum. □

Note that the utility function above is submodular and the sharing scheme is cross-monotonic. Therefore the lower bound holds for this case as well; though it is an open question whether it holds for Shapley value sharing specifically.

*Bridging the Gap between Core and Welfare.* On the positive side, we can achieve a tradeoff between approximate core stability and approximate social welfare. Towards this end, we modify Definition 17 to the following multiplicative approximation guarantee: For $\alpha \geq 1$, a feasible solution $\mathcal{F}$ to Data Exchange is $\alpha$-approximately *core stable* if there is no $U \subseteq X$ of users and another solution $\mathcal{F}'$ on the users in $U$ such that for all $i \in U$, $u_i(\mathcal{F}') > u_i(\mathcal{F})/\alpha$.

Consider the welfare optimal solution $\mathcal{F}_1$ with social welfare $W^*$ and a core-stable solution $\mathcal{F}_2$. Suppose we take a convex combination of these two solutions where we set $z_{iS} = \beta x_{iS} + (1 - \beta)y_{iS}$, where $\{x_{iS}\}$ and $\{y_{iS}\}$ denote the variables in $\mathcal{F}_1$ and $\mathcal{F}_2$ respectively. Clearly, this solution is feasible, and its social welfare is at least $\beta \cdot W^*$. Further, it is easy to check that it is $\frac{1}{1-\beta}$-approximately core stable. This shows the following theorem:

**Theorem 21.** *For any $\beta \in (0, 1]$, there is a solution $\mathcal{F}$ to Data Exchange that is simultaneously a $\frac{1}{\beta}$ approximation to social welfare and $\frac{1}{1-\beta}$-approximately core stable.*

This shows the following corollary via the Greedy rule in Appendix D.2:

**Corollary 22.** *For any $\beta \in (0, 1]$, there is a poly-time computable solution $\mathcal{F}$ to Data Exchange that is simultaneously a $O\left(\frac{\log n}{\beta}\right)$ approximation to social welfare and $\frac{1}{1-\beta}$-approximately 2-stable.*

## D.4 Strategyproofness

The discussion so far has ignored strategic considerations on the part of the agents. We now present some negative and positive results for this aspect.

*Model for strategic behavior.* We first present the model for strategic behavior. We assume that agents can choose to hide tasks or data from the clearinghouse. For agent $i$, let $u_i, h_{ij}$ denote the true utilities and shares, while $\tilde{u}_i, \tilde{h}_{ij}$ denote the reported utilities and shares. If agent $i$ reports fewer tasks, this corresponds to agent $i$ reporting $\tilde{u}_i(S) \leq u_i(S)$ for some subsets $S \subseteq X \setminus \{i\}$; correspondingly $\tilde{h}_{ij}(S) \leq h_{ij}(S)$. Since there is a centralized entity that does model refinement, if we assume agent $i$ only gets the refined model back and not any data, then the agent does not get utility for the

tasks it did not report, so that its perceived utility will be measured using $\tilde{u}_i$.

On the other hand, if agent $i$ hides data, this changes the utility perceived by other agents. For another agent $j$, we must have $\tilde{u}_j(S) \leq u_j(S)$ and $\tilde{h}_{ji}(S) \leq h_{ji}(S)$ for $S$ such that $i \in S$. Note that it could be that $\tilde{h}_{jk}(S) \geq h_{jk}(S)$ if $i \in S$, but $j, k \neq i$.

Formalizing this, let $\theta' = \{\tilde{u}, \tilde{h}\}$ be reported utilities and shares and let $\theta = \{u, h\}$ be the true values. We say that a misreport by agent $i$ is *feasible* if for the resulting $\theta'$, we have:

(1) $\tilde{u}_j(S) \leq u_j(S)$ for all $j, S$;
(2) $\tilde{h}_{jk}(S) \leq h_{jk}(S)$ when either $j = i$ or $k = i$;
(3) $\tilde{u}_j(S) = u_j(S)$ if $j \neq i$ and $i \notin S$; and
(4) $\tilde{h}_{jk}(S) = h_{jk}(S)$ if $j, k \neq i$ and $i \notin S$.

Let $\mathcal{A}$ denote an algorithm for which $\tilde{U}_i$ is the utility perceived by $i$ (measured using the utility function $\tilde{u}_i$) when $\mathcal{A}$ is implemented with misreport $\theta'$, and $U_i$ is the corresponding utility (measured using the utility function $u_i$) when $\mathcal{A}$ is run using its true report $\theta_i$. We say that $\mathcal{A}$ is *strategyproof* if for every feasible misreport by $i$, we have $\tilde{U}_i \leq U_i$.

*Approximate welfare maximizers.* We first show the strategyproofness is incompatible with welfare to any approximation. We will restrict to the class of strategyproof algorithms $\mathcal{A}$ that are *non-wasteful*, meaning that for any instance $\sigma$, if $\vec{x^\sigma}$ is the allocation found by $\mathcal{A}$, then for some agent $j$, $\sum_S x_{jS}^\sigma = 1$. In other words, the allocation cannot be scaled up while retaining feasibility.

**Theorem 23.** *For the symmetric weighted setting with proportional sharing, any non-wasteful and strateyproof algorithm $\mathcal{A}$ cannot approximate social welfare to better than a factor of $\Omega(\sqrt{n})$.*

**Proof.** We use the same instance as in the proof of Theorem 20. We follow the same proof. Note that for agents $\{1, n\}$, the only non-wasteful solution sets each of their utilities to $\sqrt{M}$. Suppose there were a strategyproof algorithm $\mathcal{A}$ that gives utility $U_1$ to agent 1 and $n$. If $U_1 < \sqrt{M}$, agent $n$ will report $s_{n-1n} = 0$, thereby killing the long cycle. But then, since $\mathcal{A}$ is non-wasteful, it will then give utility $\sqrt{M}$ to agent $n$. Therefore, any strategyproof and non-wasteful algorithm $\mathcal{A}$ has $U_1 \geq \sqrt{M}$. Continuing with the proof of Theorem 20 shows a gap of $\Omega(\sqrt{n})$ between the welfare of any such algorithm and the optimal social welfare. □

*The Greedy Cycle Canceling Algorithm.* We now present a truthful algorithm that generalizes greedy matching. Given any directed cycle $C$ on a subset of agents, let $u_C$ denote the maximum utility that can be derived if agents trade along the cycle. In other words, suppose the cycle has agents $v_1 \rightarrow v_2 \rightarrow \cdots \rightarrow v_k \rightarrow v_1$. Then we compute quantities $\vec{x} = \{x_{12}, x_{23}, \ldots, x_{k1}\} \in [0, 1]^k$, where $x_{ii+1} = x_{i+1\{i\}}$. These quantities satisfy balance:

$$x_{12}\cdot u_2(\{1\}) = x_{23}\cdot u_3(\{2\}) = x_{34}\cdot u_4(\{3\}) = \cdots x_{k1}\cdot u_1(\{k\}) = \lambda(\vec{x}).$$

We define $u_C = \max_{\vec{x}} \lambda(\vec{x}) = \min\{u_2(\{1\}), u_3(\{2\}), \ldots, u_1(\{k\})\}$.

The greedy cycle canceling algorithm works as follows: Find the directed cycle $C_1$ with largest $u_C$. Trade along this cycle and delete these agents. Again find the cycle $C_2$ with largest $u_C$ among the remaining agents, trade along it, and repeat.

THEOREM 24. *For arbitrary monotone utility functions on the agents, the* GREEDY CYCLE CANCELING *algorithm is strategyproof.*

PROOF. Focus on an agent $i$. Consider the setting with true reports, and let agent $i$ be removed in the $k^{th}$ iteration of GREEDY. If agent $i$ misreports, then it does not affect $\tilde{u}_\ell(\{\ell'\})$ for agents $\ell, \ell' \neq i$, and cannot increase $\tilde{u}_\ell(\{\ell'\})$ if either $\ell$ or $\ell'$ is equal to $i$. Therefore, $\tilde{u}_C$ for any cycle not containing $i$ remains the same, while that for cycles $C$ containing $i$ cannot increase. This implies the first $k-1$ cycles chosen by GREEDY remain the same when agent $i$ misreports. Since agent $i$'s utility will be measured using $\tilde{u}_i$, its utility in any cycle $C$ that it participates in cannot increase, since $\tilde{u}_C$ did not increase. This means agent $i$'s utility cannot increase by misreporting, showing the algorithm is strategyproof. □

As a corollary, it is immediate that the GREEDY maximal weight matching algorithm from Appendix D.2 is strategyproof. Theorems 19 and 24 together show that if trades are restricted to be among disjoint pairs of agents, then the greedy maximal weight matching is *simultaneously* core-stable, strategy-proof, and a 2-approximation to optimal welfare (in this case, the maximum weight matching).

