# OpenReview forum: "Data Exchange Markets via Utility Balancing"
_ACM.org/TheWebConf/2024/Conference — TheWebConf24_

### Official Review · Reviewer_9Wsv · 2023-11-15

**Novelty:** 4
**Technical Quality:** 3

**Review:**

Summary: The paper explores the design of a balanced data-sharing marketplace catering to entities with diverse datasets and machine learning models. This marketplace facilitates refining models using data contributed by other agents while aiming to maximize total interim social utilities, subject to constraints ensuring equitable utility contributions and receptions during model refinement.

Overall, the problem outlined in the paper is intriguing and practical, offering both theoretical and practical value. A cursory review of the proofs indicates their soundness. However, the techniques employed in the paper adhere to standard existing methodologies, and I find it challenging to discern the novelty from a technical perspective.

A few suggestions:

1) For constraints (1) and (2): It would be beneficial to specify the range of values that S should take, similar to Objective (3). The same for the summation in Theorem 10.

2) The idea of guessing-and-checking appears in various contexts before, such as makespan minimization and load scheduling. I recommend citing and discussing a few related papers in the Related Works section.

**Questions:**

My main query concerns the optimality of the approximation ratio of O(log n). Can you demonstrate that surpassing this bound is impossible unless P=NP? Establishing this would significantly strengthen the results presented in the paper.

**Reviewer Confidence:**

3: The reviewer is confident but not certain that the evaluation is correct

**Scope:**

3: The work is somewhat relevant to the Web and to the track, and is of narrow interest to a sub-community

---

### Official Review · Reviewer_uYjb · 2023-11-16

**Novelty:** 3
**Technical Quality:** 4

**Review:**

In this paper, the authors explore the challenge of incentivizing participants for data sharing and propose an approach called utility balancing to address this data exchange problem. They present a formal model for the data exchange issue, along with a utility function and a social-welfare algorithm to tackle the problem.

Throughout the paper, the authors refer to ML-based agents without providing a clear definition. It is necessary to elaborate on these agents and how they collaborate in exchanging data on behalf of participants.

The introduction section covers various topics, including incentivizing data sharing, data privacy, computational efficiency (reducing loss in refined models), and enhancing dataset utility. It is necessary to specify the specific challenges in addressing these issues and the technical hurdles involved. Additionally, it is required to clarify the concept of "loss in the refined model" – what kind of loss does it refer to.

The related work section explains the relationship between the house allocation problem and the data market problem addressed in this paper. However, it is unclear why these problems align. What existing solutions are available for the house allocation and market clearing problems, and why might reusing these solutions face limitations in solving the data sharing issue?

To maintain utility balance, there likely needs to be an enforcement mechanism or penalties associated with the proposed algorithm. It is necessary to clarify the scope of utility balance maintenance. Is it confined to participants within the same service domain or a specific region? How are these domains defined, and how is geographical vicinity or dataset similarity clusters identified for ensuring utility balance?

It is needed to elaborate on the concept of "accuracy of a task". How is this metric measured, and why is it pivotal? What's the rationale behind an agent's accuracy improving as it obtains more data from others? Additionally, it is necessary to explain why the social-welfare algorithm is an effective solution to this problem.

The evaluation lacks explanations regarding the no-sharing baseline and the matching benchmark, along with their relevance in evaluating the proposed approach. Strengthening this section is essential to substantiate the effectiveness of the proposed approach.

**Questions:**

What existing solutions are available for the house allocation and market clearing problems, and why might reusing these solutions face limitations in solving the data sharing issue?

Is the scope of utility balance confined to participants within the same service domain or a specific region? How are these domains defined, and how is geographical vicinity or dataset similarity clusters identified for ensuring utility balance?

What's the rationale behind an agent's accuracy improving as it obtains more data from others?

**Reviewer Confidence:**

2: The reviewer is willing to defend the evaluation, but it is likely that the reviewer did not understand parts of the paper

**Scope:**

2: The connection to the Web is incidental, e.g., use of Web data or API

---

### Official Review · Reviewer_n4CB · 2023-11-19

**Novelty:** 5
**Technical Quality:** 5

**Review:**

The paper presents a balanced data-sharing marketplace for entities with different datasets and machine-learning models. The marketplace aims to encourage participation and equitable utility for participants. Experiments demonstrate the effectiveness of the model.

**Questions:**

More experiments should be carried out for evaluation.

**Reviewer Confidence:**

1: The reviewer's evaluation is an educated guess

**Scope:**

2: The connection to the Web is incidental, e.g., use of Web data or API

---

### Official Review · Reviewer_KPMP · 2023-11-24

**Novelty:** 6
**Technical Quality:** 6

**Review:**

The papers study a balanced data-sharing marketplace for entities with heterogeneous datasets and machine learning models that can be refined by using data from other agents. The assumptions are that agents have diverse ML models that can be improved with data from other agents, and each agent possesses data that may be relevant to the tasks of other agents. The authors study computational complexity and existence results for the Data Exchange Problem under natural utility functions and how that utility is shared among contributors. They formally model the problem, and provide NP-hardness results. They also provide algorithms with provable guarantees and provide some simulations on a road network data.

The paper is generally well written. The problems are well formulated. The theorems seem solid and relevant. The results are interesting.
The experiments section seem a little weak though. My only concern is in the practical applicability of such a work, given the required conditions and assumptions on the utilities and marketplace.

**Questions:**

Could the authors comment on the strict requirement for the BALANCE condition? What is the practical implication of such a condition?
How can this be verified to hold in a practical setting? If it does not hold, are the results and algorithms inapplicable?

**Reviewer Confidence:**

3: The reviewer is confident but not certain that the evaluation is correct

**Scope:**

4: The work is relevant to the Web and to the track, and is of broad interest to the community

---

### Official Review · Reviewer_w7uN · 2023-12-01

**Novelty:** 6
**Technical Quality:** 5

**Review:**

The paper presents a thorough theoretical exploration of data exchange markets, proposing a novel marketplace design based on the idea that agents possess heterogeneous datasets and machine learning models. In this marketplace, agents derive utility—primarily in the form of enhanced accuracy of their machine learning models—from accessing other agents' data. This utility is then allocated to participants based on either the Shapley value or the proportional value.

Under this design, the authors address the complex problem of identifying data exchange distributions that maximize social welfare. Their comprehensive theoretical analysis establishes the NP-hardness of this problem, introduces polynomial-time approximation algorithms, and demonstrates the existence of core stable and strategy-proof solutions. Through simulations modeled on a road-sharing network, the paper shows that their approximation algorithm outperforms both a no-sharing baseline and a pairwise trade benchmark.

While the problem addressed is of critical importance and the theoretical results look solid (though beyond my area of expertise to validate the proofs), the paper's suitability for the Web conference audience is a concern. Despite its theoretical strengths, the empirical component is less convincing, especially given the significant leap from road sharing to data sharing scenarios. The paper might be better received by communities such as the EC conference rather than the broader audience of the Web conference.

**Questions:**

Clarifications on how the road sharing scenario mimics data sharing scenarios be useful. Are there more realistic data (distribution of data ownerships and/or utilities of data access) that can be used instead?

**Reviewer Confidence:**

2: The reviewer is willing to defend the evaluation, but it is likely that the reviewer did not understand parts of the paper

**Scope:**

3: The work is somewhat relevant to the Web and to the track, and is of narrow interest to a sub-community

---

### Decision · Program_Chairs · 2024-01-22

**Decision:**

Accept

**Comment:**

Summary: The paper is a theoretical exploration of data exchange markets that maximize social welfare, where agents possess heterogeneous datasets and machine learning models.

 Strengths:
 + interesting problem
 + solid theoretical results
 + well-written

 Weaknesses:
 - concern about relevance to the Web (addressed to some extent in rebuttal)
 - practical applicability as much is theoretically assumed

 Recommendation: Accept. Well-executed paper. Web relevance can be further accentuated.